# FoxO Transcription Factors: Applicability as a Novel Immune Cell Regulators and Therapeutic Targets in Oxidative Stress-Related Diseases

**DOI:** 10.3390/ijms231911877

**Published:** 2022-10-06

**Authors:** Mi Eun Kim, Dae Hyun Kim, Jun Sik Lee

**Affiliations:** 1Department of Life Science, Immunology Research Lab, BK21-plus Research Team for Bioactive Control Technology, College of Natural Sciences, Chosun University, Dong-gu, Gwangju 61452, Korea; 2LKBio Inc., Chosun University Business Incubator (CUBI) Building, Dong-gu, Gwangju 61452, Korea

**Keywords:** FoxOs, immune cells, signaling pathway, cytokines, chemokines

## Abstract

Forkhead box O transcription factors (FoxOs) play an important role in maintaining normal cell physiology by regulating survival, apoptosis, autophagy, oxidative stress, the development and maturation of T and B lymphocytes, and the secretion of inflammatory cytokines. Cell types whose functions are regulated by FoxOs include keratinocytes, mucosal dermis, neutrophils, macrophages, dendritic cells, tumor-infiltrating activated regulatory T (Tregs) cells, B cells, and natural killer (NK) cells. FoxOs plays a crucial role in physiological and pathological immune responses. FoxOs control the development and function of Foxp3+ Tregs. Treg cells and Th17 cells are subsets of CD4+ T cells, which play an essential role in immune homeostasis and infection. Dysregulation of the Th17/Treg cell balance has been implicated in the development and progression of several disorders, such as autoimmune diseases, inflammatory diseases, and cancer. In addition, FoxOs are stimulated by the mitogen-activated protein (MAP) kinase pathway and inhibited by the PI3 kinase/AKT pathway. Downstream target genes of FoxOs include pro-inflammatory signaling molecules (toll-like receptor (TLR) 2, TLR4, interleukin (IL)-1β, and tumor necrosis factor (TNF)-α), chemokine receptors (CCR7 and CXCR2), B-cell regulators (APRIL and BLYS), T-regulatory modulators (Foxp3 and CTLA-4), and DNA repair enzymes (GADD45α). Here, we review the recent progress in our understanding of FoxOs as the key molecules involved in immune cell differentiation and its role in the initiation of autoimmune diseases caused by dysregulation of immune cell balance. Additionally, in various diseases, FoxOs act as a cancer repressor, and reviving the activity of FoxOs forces Tregs to egress from various tissues. However, FoxOs regulate the cytotoxicity of both CD8+ T and NK cells against tumor cells, aiding in the restoration of redox and inflammatory homeostasis, repair of the damaged tissue, and activation of immune cells. A better understanding of FoxOs regulation may help develop novel potential therapeutics for treating immune/oxidative stress-related diseases.

## 1. Introduction

Forkhead box O transcription factors (FoxOs) are known to play an important role in regulating the immune and inflammatory responses of the human body against various infections [1,2], neurodegenerative diseases [3,4], and metabolic disorders and diseases, such as obesity, type 2 diabetes mellitus, and non-alcoholic fatty liver disease (NAFLD) [5]. The evolutionally conserved FoxO family consists of FoxO1, FoxO3, FoxO4, and FoxO6 in mammals [6]. These FoxOs are characterized by a highly conserved DNA binding motif, known as a forkhead box O or a winged helix domain, and it regulates various downstream target genes involved in the cell cycle, cell death, and oxidative stress response [7,8]. FoxOs are expressed in the ovaries, prostate, skeletal muscle, brain, heart, lung, liver, pancreas, spleen, thymus, and testes [9,10]. One of the key regulatory mechanisms of FoxO factors involves the phosphorylation reaction. Phosphorylated FoxOs by protein kinase B in response to insulin or several growth factors (PKB, also known as Akt), are allowed to translocate from the nucleus to the cytoplasm [7,8]. Another regulatory step involves the acetylation of FoxO as a posttranslational modification. The cAMP-response element-binding protein (CREB)-binding protein (CBP) triggers the transactivation function of FoxOs, whereas it leads to the attenuation of their transcriptional activity following the acetylation [11,12]. Normally, FoxOs upregulate various pro-inflammatory cytokines, such as interleukin (IL)-1β, IL-9, toll-like receptor (TLR)1, and TLR4, which not only modulate the host inflammatory reaction, but also alter the innate immune response [13,14]. Moreover, FoxOs are essential for adaptive immune functions, including maturation and differentiation of B and T lymphocytes [15,16]. Thus, the interplay of FoxOs with immune cells presents a possible vital coalition that can be targeted to tackle detrimental inflammation. The FoxO family of transcription factors plays a profound role in regulating the development and functioning of immune cells [17,18,19]. Additionally, it exerts immunoregulatory effects by modulating non-malignant cells and non-immune cells. FoxOs are also rendered inactive in diseased tissue relative to its normal counterpart, such as idiopathic pulmonary fibrosis (IPF) [20,21]. FoxOs play an important role in regulating several aspects of mucosal immunity by affecting dendritic cells (DCs) [22], recruiting and activating macrophages and neutrophils [23,24,25], and aiding in the development and functioning of T helper (Th) cells and B-lymphocytes [26,27,28]. In the case of FoxO1, it affects immune responses by controlling cytokine production [29] and protecting hematopoietic stem cells (HSCs) from oxidative stress [30]. FoxO1 is activated by bacteria in DCs and promotes DC phagocytosis, migration, homing to lymph nodes, stimulation of T cells, B cell activation, and antibody production [31]. Phagocytosis by hemocytes is an important mechanism for cellular immunity against pathogenic infection [32]. FoxO1-mediated autophagy is required for natural killer (NK) cell development [33]. FoxO is also involved in promoting bacterial phagocytosis by neutrophils [25]. In addition, FoxO1 regulates important aspects of keratinocyte function and potentially plays a role in maintaining or repairing the epithelial barrier [34,35]. Moreover, FoxO3 is known to control the magnitude of T cell immune responses by modulating dendritic cell functions [36]. FoxO3-deficient DCs sustain T cell viability by producing increased levels of IL-6. In addition, CTLA-4-Ig-mediated stimulation of DCs induces the nuclear localization of FoxO3, which in turn inhibits IL-6 and tumor necrosis factor (TNF) production. Based on these results, it has been concluded that FoxO3 is important for regulation of immune cells [36].

Several studies have reported that kaempferol caused a translocation of the *C. elegans* FoxO1 factor [37], and epigallocatechin gallate (EGCG) prevented the development of cardiac hypertrophy through reactive oxygen (RS)-dependent and -independent mechanisms [38]. Further, the antiangiogenic effects of EGCG arose through the activation of FoxO by inhibiting PI3K/Akt [39]. However, baicalin treatment suppressed systemic inflammatory stress by reducing serum TNFα levels, and a significant reduction in serum insulin and glucose levels resulted in ameliorated insulin resistance [40]. Therefore, this review underscores the importance of FoxO proteins in the mechanistic regulation of host inflammatory and immunological responses.

## 2. Roles of the FoxO Family

FoxO proteins are expressed in nearly all tissues. They act as regulators of pleiotropic functions within cells, which have considerable consequences for host health and disease [41]. FoxOs transcriptionally modulate the expression of a multitude of downstream effector genes involved in cellular proliferation, cell cycle arrest, apoptosis, genomic repair, metabolic balance, redox homeostasis, and resistance to oxidative stress [42]. Sequential phosphorylation of FoxOs by the phosphoinositide-3-kinase/protein kinase B (PI3K/Akt) pathway [43] in the presence of growth factors results in cytoplasmic sequestration or ubiquitination, thus rendering them inactive. In the absence of growth factors, phosphatase and tensin homologue (PTEN) abolishes PI3K/Akt-mediated phosphorylation of FoxOs, thus leading to their dephosphorylation and subsequent nuclear shuttling. Once in the nucleus, FoxO factors are involved in transcriptional regulation of several downstream target genes [44,45]. However, disruption of FoxOs results in involution of the IGF1R pathway, which prolongs communication between macrophages and B cells and, under the condition of insufficient T cell feedback, permits the production of IgM that targets the canonical auto-antigens dsDNA, Fc-portion of IgG, and cyclic citrullinated peptides [46].

FoxO proteins are primary regulators of the innate immune system [2]. This is exemplified by the management of inflammation by FoxOs through escalated TLR3/4- mediated signaling and IL-1β expression in human macrophages [1]. FoxO1 stimulates the transcriptional expression of pro-inflammatory molecules such as TLR1 and 4, IL-1β, and TNF-α; chemokine receptors such as C–C chemokine receptor type 7 (CCR7) and C-X-C chemokine receptor type 2 (CXCR2); B cell modulators such as APRIL (a proliferation inducing ligand) and BLYS (B lymphocyte stimulator); and T cell regulators such as CTLA-4 (cytotoxic T-lymphocyte-associated protein 4), in addition to stimulating antioxidants. FoxO1 is also essential for T-cell tolerance and naive T-cell homeostasis, homing of DCs and B cells, and initiation of an adaptive immune response to bacterial challenges [14,41]. Furthermore, FoxO1 is involved in transcriptional modulation of IL-9-generating Th 9 cells, which participate in inducing immunity against extracellular pathogens. Intriguingly, pulmonary overexpression of IL-9 has been shown to play a role in lymphocytic and eosinophilic inflammatory infiltration, mast cell hyperplasia, and mucus secretion [47]. Notably, FoxO3 is expressed within the airway epithelium in addition to macrophages and other cell types in the lungs [48,49]. FoxO3 plays an extremely important role in overseeing innate immune responses to infections in the airway epithelium. In response to bacterial challenges in bronchial epithelial cells, FoxO3 induces the expression of antimicrobial factors, such as human β-defensin 2, and various cytokines, including IL-6, IL-8, TNF-α, and C-X-C motif chemokine ligand 10 (CXCL10) [1]. Moreover, FoxO3 is expressed in immune cells. Hence, interest in investigating the importance of FoxO3 in lymphoid homeostasis has surged recently [27,36,50]. Upregulation of FoxO3 has been observed in polymorphonuclear cells and peripheral blood mononuclear cells in patients with rheumatoid arthritis (RA) [51]. Overexpression of FoxO3 is mediated by T cell receptor stimulation [52]. Consequently, FoxO3 promotes the polarization of CD4+ T cells towards pathogenic Th cells, producing interferon γ and granulocyte monocyte colony-stimulating factors. FoxO3−/− mice exhibit reduced susceptibility to experimental autoimmune encephalomyelitis [52]. Both FoxO1 and FoxO3 have largely redundant but complex roles in maintaining T-cell quiescence and in controlling the response to growth factors and inflammatory stress [53,54]. Loss of FoxO1 in T cells results in the development of a mild lymphoproliferative and autoimmune phenotype [27,55]. This phenotype is distinct from that of mice with regulatory Treg-specific deletion of FoxO1, in which lethal inflammation is observed after the loss of dominant tolerance without compromising conventional T-cell function [56].

## 3. Function of the FoxO Transcription Factors in Immune Cells

### 3.1. Hematopoietic Stem Cells (HSCs)

FoxO proteins are essential for proper functioning and regulation of stem cells in multiple adult tissues [57]. These include hematopoietic, neural, and muscle stem cell pools [58,59,60,61]. In this review, we have focused on the role of FoxOs in HSCs. FoxO proteins are essential for maintaining the HSC pool and activity [59,62,63]. Conditional deletion of FoxO1/3/4 or deletion of FoxO3 alone compromised the HSC pool and long-term repopulation capacity of HSCs in mice [59,62,63,64,65]. In addition, FoxO1 and FoxO3 are the main FoxO factors expressed in HSCs, and the regulation of subcellular localization and activity of FoxO1 and FoxO3 is relatively distinct [66,67]; FoxO1 has a more cytosolic localization, and FoxO3 is almost entirely localized to the nucleus in HSCs with long-term repopulation ability [62,66,67,68]. Taken together, these results suggest that FoxO3 is the main active FoxO protein in HSCs. In addition to compromised HSC function, loss of FoxO3 leads to increased myeloproliferation, anemia, and immune deficiencies similar to those observed in triple FoxO-deleted mice [59,62,63,69,70,71]. As such, among the FoxO family members, FoxO1 and FoxO3 play an important role in HSCs in addition to playing an important role in immune functions.

### 3.2. Dendritic Cells

FoxO1 is generally found in its phosphorylated, inactive form in DCs, which ensures their survival and normal proliferation, especially when DCs are stimulated by CCR7 ligands or immunological synapse formation with T cells [72,73]. Knockdown of FoxO1 under serum-free conditions leads to a significantly lower percentage of apoptotic DCs [73]. The transcriptional activity of FoxO also plays a fundamental role in maintaining normal homing of DCs to lymph nodes. In particular, FoxO1 promotes the transcription of CCR7 and intercellular adhesion molecule-1 (ICAM-1), both of which are critical for the DC-mediated stimulation of T and B lymphocytes in response to bacterial infection [22,74,75]. In contrast, FoxO3 suppresses DC production of key inflammatory cytokines, such as IL-6 and TNF, and constrains CD4+ and CD8+ T-cell population expansion after viral infection [36]. FoxO3-deficient dendritic cells produce high levels of IL-6, which sustains T-cell viability and expansion in response to lymphocytic choriomeningitis virus (LCMV) infection [36]. Therefore, FoxOs may be different or opposite among different cell types (immune vs. non-immune cells).

### 3.3. Macrophages

The role of the FoxO pathway in the immune system, particularly with regard to macrophages, remains controversial. Some researchers have reported that FoxO3 is predominantly expressed in myeloid cells, including macrophages, whereas others have not detected the expression of either FoxO1 or FoxO3 [76,77]. Interestingly, in macrophages, FoxO1 has been found to stimulate both pro- and anti-inflammatory pathways by upregulating the TLR4 and IL-10 promoters, respectively [23,24]. Simultaneous deletion of FoxO1/3/4 induces monocytosis, increased NOS2 (iNOS) expression, and oxidative stress in mice [78].

During classical activation following lipopolysaccharide (LPS) treatment, M2-like macrophages showed increased expression of FoxO1 compared with M1-like macrophages. Furthermore, FoxO1 tends to bind to the IL-10 promoter in M2-like macrophages but not in M1-like macrophages. After a challenge with LPS, lysozyme 2 (LysM)-Cre mice, with macrophage-specific deletion of FoxO1, showed a reduction in M2-like cells and an increase in M1-like cells. This could be deduced to be due to a significant reduction in the expression of M2-like macrophage markers (IL-10, Arg1, Fizz1, and IL-13 receptor alpha 1 (IL-13Rα1)), and increased expression of M1-like macrophage markers (inducible iNOS, IL-12α, IL-12β, and CCR2) compared with that of the wild-type [23].

### 3.4. T Cells and B Cells

T and B cells express FoxO1 and FoxO3, respectively. FoxOs are critical for T cell homeostasis [18]. T-cells homeostasis is an important cellular process defined by the ability of the immune system to maintain normal T-cell counts and, at the same time, replenish the T-cell counts following T-cell depletion or expansion [79]. Although FoxOs have been shown to promote apoptosis in response to nutrient or cytokine withdrawal in lymphocytes, their exact role in T cells remains complex. Kerdiles et al. showed that the conditional deletion of FoxO1 alters T-cell homeostasis [53]. FoxO1 is essential for the regulation of several genes involved in T-cell trafficking and survival. Furthermore, FoxO1 is involved in the negative feedback regulation of growth factor signaling, coupled with homing of naive T cells and their subsequent survival [53,80]. Another study showed that the expression of constitutively active FoxO1 in Jurkat cells led to the transcriptional activation of genes involved in lymphocyte recruitment into secondary lymphoid organs [81]. FoxO1 also plays a critical role in the differentiation of memory CD8+ T cells. A diverse array of studies have highlighted the essential role of FoxO in regulating specialized lymphocyte functions. FoxO1 inactivation directs the homeostasis of CD4+ conventional and Treg cells, whereas enforced expression of FoxO1 inadvertently hampers this balance [82]. The abrogation of FoxO expression was also linked to a progressive decrease in the frequency of Tregs in peripheral tissues, and their immune-suppressive capacity was found to be significantly hampered, thus emphasizing the importance of FoxO proteins in maintaining immunological tolerance [83]. Tregs specifically depleted of FoxO1 produce more IFN-γ than the wild-type cells do [84]. This is in agreement with previous observations of IFN suppression by FoxO proteins. Utzschneider et al. found that the continued expression of FoxO1 is indispensable for preserving longevity, self-renewal, and the ability to shift between quiescence and cell division of the CD8+ memory T-cell population [15]. Inactivation of FoxO1 leads to the reversion of memory T cells to a state of terminal differentiation, which prevents a secondary memory response in multiple cases of infection [85]. Deletion of FoxO1 after the clearance of an infection resulted in a rapid loss of typical gene expression patterns in memory T cells. Even during a persistent viral infection, the depletion of FoxO1 activity caused a dramatic decline in T-cell expansion, while giving rise to T cells deficient in effector cytokines and exhibiting features of anergy [15,84,86]. This underscored the broad importance of FoxO1 for manifesting the post-effector immune program, a prerequisite for forming the long-lived memory of T cells. Despite the increased expansion of Foxo3-deficient effector T cells, precursors of memory T cells also accumulate, resulting in an increased quantity of CD8+ memory T cells. However, this increase in CD8+ memory T cells does not trigger a stronger recall response, suggesting that FoxO3 may also function in regulating the recall responses of memory T cells [50].

FoxO1 is also important for the proliferation, differentiation, survival, and class switching of B cells. FoxO1 has been shown to direct the development of germinal centers, which are necessary for the development of clonal variants of B cells. The depletion of FoxO1 in germinal center B cells led to diminished somatic hypermutation and dwindled class switching, which significantly hampered a robust antibody response to infections [87]. The loss of FoxO1 in DCs results in the reduction in occurrences of multiple phenomena, such as cytokine production, homing of DCs to the lymph nodes, activation of CD4+ T and B cells, and antibody generation, thereby enhancing the sensitivity to pathogenic challenges [31]. FoxO3 was also identified as a prime modulator of CD8 T-cell memory, and FoxO3 therapeutic modifications have been proposed to convalesce protective immunity against intracellular pathogens [50]. A deficiency in FoxO3 following a viral infection has been shown to facilitate considerably exaggerated expansion of T-cell populations. This is due to the DC-specific increase in the production of IL-6. This causes variations in the stimulatory capacity of FoxO3-deficient DCs to sustain T-cell viability. The use of CTLA-4-Ig-mediated stimulation led to FoxO3 nuclear localization, which consequently suppressed the heightened release of IL-6 and TNF. These data suggest that FoxO3 contributes to the production of key inflammatory cytokines and controls T-cell viability [36]. FoxO1 is critical for class switch recombination, which mediates antibody diversity in B cells. Loss of FoxO1 leads to decreased immunoglobulin heavy chain production, concomitant with decreased expression of B-cell-specific activation-induced cytidine deaminase, which initiates class switch recombination [26,88]. These studies demonstrate the importance of FoxO1 and FoxO3 in T and B cell biology.

### 3.5. Natural Killer (NK) Cells

NK cells, a major component of innate immunity, serve as the first line of defense against transformed tumors and virus-infected cells [89,90]. NK cells were recently defined as part of group 1 innate lymphoid cells, according to their cytokine secretion patterns [91]. Cytokine secretion and granule-mediated cytotoxicity are the two major effector functions of NK cells that are critical for early immune responses [90,92]. Similarly to leukocyte populations, NK cells are also derived from HSCs in the bone marrow (BM). Each step in NK cell development is finely regulated via signaling by various cytokines and transcription factors. A recent study showed that FoxO3a triggers autophagy, which is essential for lifelong maintenance of HSCs [93]. FoxO1-induced autophagy was shown to be indispensable for NK cell development and murine cytomegalovirus (MCMV) clearance using an NKp46-Cre mouse model [33]. Deng et al. reported that FoxO1 is dispensable for NK cell development and that inactivation of FoxO1 is required for T-bet expression [94]. Furthermore, Luu et al. reported that the combined loss of FoxO1 and FoxO3 caused specific changes in the composition of noncytotoxic innate lymphoid cell subsets in the BM, thymus, and spleen [95]. They also revealed that FoxO transcription factors ensure proper NK cell development at various lineage commitment stages by orchestrating distinct molecular mechanisms. Combined FoxO1 and FoxO3 deficiency in common and innate lymphoid cell progenitors results in reduced expression of genes associated with NK cell development [96].

### 3.6. Potential Considerations for FoxO Proteins in Various Diseases

The FoxO family members, in addition to acting as sensors for oxidative stress signals, also act as regulators of subsequent cellular responses. The transcriptional network downstream of these redox-sensitive proteins is at least partially dependent on oxidative status. Kops et al. demonstrated that FoxO facilitates the synthesis of reactive oxygen species (ROS)-scavenging enzymes, such as manganese superoxide dismutase (MnSOD) and catalase, in response to intercellular oxidative stress [97]. Activation of FoxO3 results in the destabilization of hypoxia-inducible factor (HIF)-1α and suppresses hypoxia-mediated increases in ROS [98]. A plethora of evidence points to the fact that insufficient FoxO activity may cause elevated cellular damage in the presence of high concentrations of ROS.

#### 3.6.1. Role of FoxOs in Autoimmune and Inflammation

Excessive oxidative stress is known to reduce type I and type III IFN responses to viral infection in airway epithelial cells [99]. In addition, strong nuclear staining for FoxO3 is found in the lungs of patients with a variety of infection-related lung disorders, including cystic fibrosis, chronic obstructive pulmonary disease, and severe pneumonia with acute respiratory distress. In such individuals, FoxO3 levels are negatively correlated with IL-8 production in airway epithelial cells [48]. In addition, FoxO3 blocks oxidative stress, thereby suppressing lung inflammation in mice exposed to cigarette smoke [49]. These results underscore the contribution of FoxO3 to both the regulation of antiviral responses and inhibition of pro-inflammatory chemokine expression. By potentially reducing the expression of inflammatory cytokines in response to viral infections, FoxO3 activation provides protection against lung inflammation. Lin et al. showed that ablation of FoxO3 may lead to spontaneous lympho-proliferation, T-cell hyperactivation, and escalated inflammation with a pronounced increase in the levels of inflammation-favoring molecules, such as NF-κB, IL-2, and IFN-γ [100]. FoxO proteins participate in the antibacterial and antiviral innate immune responses of invertebrates [101,102]. Chronic activation of FoxO in aged Drosophila suppresses the expression of the peptidoglycan recognition protein SC2 (a negative regulator of the immune deficiency (IMD) pathway) and disrupts intestinal immune homeostasis [103]. FoxO3 plays an important role in improving symptoms of glucocorticoid-mediated systemic lupus erythematosus (SLE) by inhibiting NF-κB activity [104]. Accumulating evidence indicates that FoxO may play an important role in the regulation of viral infections. For example, FoxO1 contributes to the transcription and replication of the hepatitis B virus (HBV) through the activation of the HBV core promoter [105,106]. FoxO3 acts as a negative regulator of virus-specific CD8+ T-cell responses in the chronic lymphocytic choriomeningitis virus infection, and has been proposed as a potential therapeutic target for chronic viral infections [107]. FoxO4 may also inhibit HIV-1 replication by acting as a transcriptional repressor. Oteiza et al. demonstrated that overexpression of a constitutively active FoxO4-TM mutant antagonized HIV-1 transcription reactivation in response to T-cell activators, such as PMA or TNF-α [108]. Hence, FoxO factors can control HIV-1 transcription and provide new insights into their important role during the establishment of HIV-1 latency [108].

The production of a large amount of IL-9 is vital for allergic inflammatory response, autoimmune syndrome, and immunity to pathogenic invasion in Th2, Th9, and Th17 cells [56]. FoxO1 is a critical transcription factor necessary for IL-9 induction in these immune cells. Mechanistic insights indicated that FoxO1 transactivated IL-9 in these T cells. This is the primary mechanism deployed by FoxO1 to ameliorate allergic inflammation, as observed in asthma [13]. However, much less is known about the role of FoxO1 in Th17 generation. Deficiency of FoxO1 regulates Th17 differentiation by the defective Treg cell levels [109]. FoxO1 is considered as an anti-inflammatory control switch directly acting on the Th17 program [110]. Thus, FoxO1 regulates the expression and function of PD-1, and leads to therapeutic options for chronic viral infection or cancer [111]. FoxO1 regulates memory T cell differentiation and maintenance through mammalian target of the rapamycin complex 2 (mTORC2) pathway [112]. N-Acetyl Cysteine (NAC), antioxidant capacity reduces FoxO1 through Akt activation, and leads to improved tumor control. This also extends to transduced murine T cells [113].

#### 3.6.2. Regulation of FoxOs in Inflammation-Induced Diseases

FoxO1 drives the production of IL-10 by transcriptionally monitoring its expression [114]. IL-10 is upregulated in SARS-CoV-2 patients to an appreciable extent. Considering the vital contribution of FoxO factors in maintaining a check on excessive inflammation, its clinical significance in fighting the cytokine storm induced in patients with COVID-19 is logically implied. However, FoxO1 induces transcription of the memory transcription factor EOMES and inhibits the effector transcription factor T-bet [86]. Continuous expression of FoxO1 is required to maintain the survival, renewal, and gene expression profiles of the memory subset [15]. FoxO1 stimulates the transcription of memory markers IL-7R and CD62L, which are involved in homing and long-term survival, but negatively regulates effector functions, including the production of interferon-γ and granzyme B [115,116]. The strong association between smoking and RA led us to speculate that smoking-related changes in the immune cell phenotype could contribute to the development of RA. The transcription levels of FoxO1 in peripheral blood monocytes reported lower levels of FoxO1 mRNA in RA patients than in healthy controls [117]. This result was consistent with lower FoxO1 expression observed in nicotine-stimulated cells, but not with an increased abundance of cells with naive or memory phenotypes [118]. Overexpression of FoxO3 in various diseases, such as type 1 diabetes mellitus, suggests a potential role of this gene in the development of autoimmune diseases. FoxO3 contributes to dysregulation of immune tolerance [119]. FoxO1 plays an important role in maintaining homeostasis in periodontal tissues and in response to bacterial challenges. Alterations in FoxO1 function have a significant effect on periodontal disease susceptibility because FoxO1 is involved in the regulation of leukocyte function [14]. Collectively, FoxO transcription factors regulate various immune cell diseases (Figure 1).

Glucosamine suppress lung cancer cell proliferation by affecting the transcriptional activity of FoxOs through inhibition of p27kip1 and p21cip1, which are involved in cell cycle arrest, and Bim and FasL, which are involved in apoptosis [120]. In addition, MPSSS, a novel polysaccharide purified from *Lentinus edodes*, has been reported to have anti-tumor activity [121]. *M*. *scabra* flavonoids (MSF), in which the major active compounds were luteolin, apigenin, kaempferol, and moslosooflavone, decreased protein expressions of phosphating Akt/FoxO1 in both the lung tissues of IAV-infected mice as well as IAV-infected macrophages [122]. FoxO3a played an important role in glucocorticoids treatment of SLE by suppressing pro-inflammatory response. Targeting FOXO3a might provide a novel therapeutic strategy against SLE [104]. β-catenin/FoxOs axis serves as a bridge between environmental factors and autoimmune disease by modulating T_reg_ properties [123]. The collective evidence strongly supports the theory that the effects of pharmacological compounds on various diseases are achieved as summarized in Table 1.

## 4. Conclusions

In the past decade, it has become clear that the FoxO transcription factors are key regulators of homeostatic hematopoiesis and are implicated in many fundamental processes. Future studies may reveal the conditions under which deregulated FoxO function leads to hematological disorders. These studies may particularly illuminate the influence of FoxO on haem malignancies. Identifying FoxO-regulated programs that protect HSCs from damage caused by various diseases may provide important information related to FoxO alterations that could lead to blood disorders in the elderly. FoxO transcription factors play a role in almost every aspect of T-cell biology examined so far. They respond to a wide range of extrinsic signals to fundamentally alter the trajectory of the T cell-dependent immune response. Programmed gene expression includes cell type-specific genes involved in differentiated functions, as well as genes that control the essential aspects of general cellular physiology, such as cell division, survival, and metabolism (Figure 2). The challenge is to isolate the direct effects of FoxO transcriptional regulation from the indirect effects that ripple and echo throughout the cellular signaling network. In-depth study of FoxO transcription factors in the immune system would illuminate and eventually resolve issues of medical significance. It is expected that better understanding of the modulatory mechanisms of FoxOs will provide a basis for the discovery of molecular targets that can therapeutically modulate inflammatory conditions and various diseases, and also enable the development of potential effective interventions to delay diseases.

## Figures and Tables

**Figure 1 ijms-23-11877-f001:**
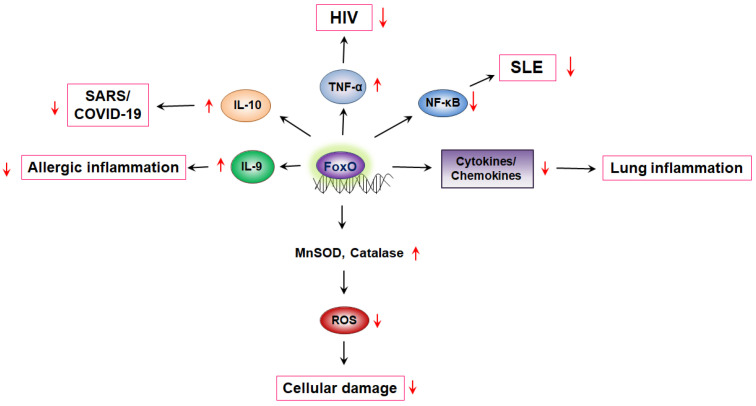
FoxO-mediated cytokine production regulates diseases. FoxO, Forkhead transcription factor O; MnSOD, manganese superoxide dismutase; ROS, reactive oxygen species; NF-κB, nuclear factor kappa B; SLE, systemic lupus erythematosus; TNFα, tumor necrosis factor alpha; HIV, human immunodeficiency virus; SARS, severe acute respiratory syndrome.

**Figure 2 ijms-23-11877-f002:**
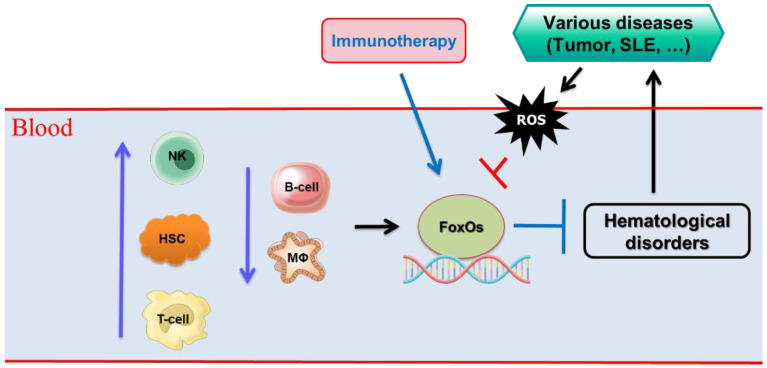
Various cell types involved in hematological disorders through FoxO transcription factors. T-cell, T lymphocyte; B-cells, B lymphocytes; NK cells, natural killer cells; HSCs, hematopoietic stem cells; MΦ, macrophages, ROS, reactive oxygen species.

**Table 1 ijms-23-11877-t001:** Effect of pharmacological compounds in various diseases.

Pharmacological Compounds	Target Genes	Diseases	References
Glucosamine	p27, p21, Bim, FasL	Cancer	[120]
MPSSS	p21	Tumor	[121]
MSF	Cytokines (TNFα, IL-6)	IAV-induced lung injury	[122]
Glucocorticoids	Interaction of NF-κB (TNFα, IL-6, mcp-1)	SLE	[104]
β-catenin	IFN-ɣ, IL-10	Autoimmune	[123]

## Data Availability

The data presented in this study are available on request from the corresponding author. The data are not publicly available due to privacy.

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
