# Peer review of "FoxO Transcription Factors: Applicability as a Novel Immune Cell Regulators and Therapeutic Targets in Oxidative Stress-Related Diseases"

_ijms, 2022, doi:10.3390/ijms231911877_

Round 1

Reviewer 1 Report

The authors begin the paper with a brief introduction to this protein family, talking specifically about its function in various immune response cells and ending with its importance in some diseases. I believe that the text addressed several important aspects of the subject and the illustrations help in understanding.

I think this will be a great contribution to the IJMS

Author Response

The authors begin the paper with a brief introduction to this protein family, talking specifically about its function in various immune response cells and ending with its importance in some diseases. I believe that the text addressed several important aspects of the subject and the illustrations help in understanding.

I think this will be a great contribution to the IJMS.

Response: Thanks for your comments.

Reviewer 2 Report

This is a Review article on the role of the Forkhead Box O family of transcription factors (FoxOs) mainly focusing on the adaptive and innate immune system. It also aims to summarize the role of FoxOs in oxidative stress-related diseases and potential use as a therapeutic target.  

The authors have gathered a lot of information (past and recent) on the subject, however, reading the manuscript gives the impression that the information is diffuse, not well organized and being repeated. The manuscript needs to be re-organized in clearly defined sections that gather all relevant information, for example everything about DCs and other cell types collected in one section (as in the section 3). In the current version, information mentioned in the introduction about various cell types appear again later in the text. There are two sections (2 and 3) with the same title “Function of the FoxO family” and so it is not clear how the authors intend to separate this information. Section 3 is the most organized one, and maybe the authors could keep this one as the main part and move all relevant information of the introduction (which can be decreased) and section 2 in that one. The title is about using FoxOs as potential therapeutic targets in oxidative stress-related diseases, however there was no background of which diseases fall into this definition in the introduction. Also, there were no information included related with the therapeutic efforts, and pharmacologic approaches for targeting the FoxO factors. The role of FoxOs (FoxO1) on Th17 has not been adequately addressed. Here is a paper on the role of FoxO factors in Th17 differentiation that could be helpful: Laine, A. et al. Foxo1 is a T cell-intrinsic inhibitor of the RORgammat-Th17 Program. J. Immunol. 195, 1791-1803 (2015). FoxO factors have also been involved in T cell metabolism, fitness, exhaustion and memory in chronic infections and cancer involving mitochondria and ROS production. (also relevant as part of this IJMS Special Issue ‘Chronic Inflammation and Related Diseases: From Mechanisms to Therapies”). Maybe the authors could add some of this information. Below are some relevant papers that could be discussed.

Staron, M.M. et al. The transcription factor FoxO1 sustains expression of the inhibitory receptor PD-1 and survival of antiviral CD8(+) T cells during chronic infection. Immunity 41, 802-814 (2014).

Scharping, N.E. et al. The tumor microenvironment represses T cell mitochondrial biogenesis to drive intratumoral T cell metabolic insufficiency and dysfunction. Immunity 45, 374-388 (2016).

Zhang, L. et al. Mammalian Target of Rapamycin Complex 2 Controls CD8 T Cell Memory Differentiation in a Foxo1-Dependent Manner. Cell Rep. 14, 1206-1217 (2016).

Odorizzi, P.M., Pauken, K.E., Paley, M.A., Sharpe, A. & Wherry, E.J. Genetic absence of PD-1 promotes accumulation of terminally differentiated exhausted CD8+ T cells. J. Exp. Med. 212, 1125-1137 (2015).

Scheffel, M.J. et al. N-acetyl cysteine protects anti-melanoma cytotoxic T cells from exhaustion induced by rapid expansion via the downmodulation of Foxo1 in an Akt-dependent manner. Cancer Immunol. Immunother. 67, 691-702 (2018).

Please see below some suggestions for improving the manuscript structure:

1. Introduction. Should include the general background about FoxO biology, members, structure and general mechanisms of action and activity regulation, not in extensive detail, but briefly by referring to previous reviews, without many details about their cell-specific function and also provide brief information about the oxidative stress-related diseases that will be discussed later in the text. In relation to their structure and activity regulation you can briefly refer to the opportunities opened to target FoxOs therapeutically (e.g. interfering with other proteins that regulate FoxO activity, nuclear transclocation, protein interactions etc…) which will be discussed in more detail later.

2. Function of the FoxO family of transcription factors. Here add the role of FoxOs in the various immune cell types, collecting all the diffuse information from the current version in clearly defined sections.

3. Current section 3.6 could become section 3 on the role of FoxOs on oxidative stress-related diseases.  Maybe here separate two bigger subsections for autoimmune/inflammatory diseases and cancer/tumor immunotherapy, with subsections on the diseases you want to focus on (e.g. allergic inflammation, SLE, lung inflammation, tumor immunotherapy etc…).

4. Therapeutic exploitation of FoxO transcription factors. Here you can add current state about pharmacological approaches targeting FoxOs focused on the above discussed diseases. Adding a summarizing table would be very helpful for the readers.

5. Conclusions. Summary of current state in targeting FoxOs therapeutically (focused on the discussed diseases), future perspectives and challenges.

The figures are nice, but I think the second figure can come first with the part mentioning the role of FoxOs in various cell types and then add the figure about FoxOs and ROS in diseases along with that part, including T cell exhaustion and role in tumor immunotherapy/TME if you decide to include it.

The first paragraph of the conclusions seems a bit odd. These details can also be moved appropriately to relevant section of the main text. Then, the conclusions can start with the second paragraph that you already have.

Lines 99-100 and 141: Opposite role of FoxO3 in epithelial versus DCs in IL-6, TNFa production. Maybe discuss that the role of FoxOs may be different or opposite among different cell types (immune vs non-immune cells).

There is a mistake in Line 64: …CTLA-4-Ig-mediated stimulation… (in ref 26, Dejean et al used CTLA-4-Ig to stimulate B7 receptors on DCs)

There are also some minor corrections for improving the text:

Title: Since FoxO includes a family of many factors (FoxO1, FoxO3…), it would be more appropriate to write as: FoxO transcription factors: Applicability as novel immune cell regulators and therapeutic targets in oxidative stress-related diseases.

Line 12: Forkhead box O transcription factors (FoxOs) play… The same modification is suggested in other places in the text.

Line 15: …by FoxOs…

Lines 16-17: …FoxOs play…responses. They control the…

Line 21: In addition, FoxOs are stimulated…

Line 23: …of FoxOs…

Line 27: …of FoxOs as key molecules…and their role…

Line 29: …FoxOs act as cancer repressors…activity of FoxOs…

Line 30: …FoxOs regulate…

Line 32: …of FoxOs regulation…

Line 37: Forkhead box O transcription factors (FoxOs) are known to play…

Line 39: Normally, FoxOs upregulate…

Line 42: …FoxOs are essential…

Line 43: …of FoxOs…

Line 47: …FoxOs are also rendered inactive…

Line 49: …FoxOs play…

Line 53: …stem cells…

Line 55: …homing to lymph nodes…

Line 59: FoxO1 is also… Publication (Dong et al. 2017) needs formatting.

Line 61: Remove “function”.

Line 72: FoxOs transcriptionally modulate…

Line 75: Sequential phosphorylation of FoxOs by…

Line 79: …of FoxOs, thus leading to their…

Line 82: FoxO proteins are primary regulators of…

Line 83: …by FoxOs…

Line 88: …CTLA-4…

Line 89: …antioxidants.

Section 2 has same title as section 3: “Function of the FoxO family”.

Line 117: FoxO proteins are…

Line 119: …in the role of FoxOs… FoxO proteins are…

Line 122: Citation reformatting needed to put all refs in one bracket: ...[47, 49] [44, 48, 50].

Line 126: …main active FoxO protein…

Line 129: Suggested modification: …among the FoxO family members…

Line 147: Macrophages

Line 168: …T cell homeostasis

Line 170: …FoxOs have…

Line 182: I think saying …hampers this balance… would be better.

Line 185: …importance of FoxO proteins in maintaining…

Line 216: …the use of CTLA-4-Ig-mediated stimulation led to…

Line 237: …using an NKp46-Cre mouse model. (remove “that differs from the model”)

Line 239: There is an extra space after [79].

Line 270:…FoxO proteins participate…

Line 284: …TNF-α

Figure 1: Suggested modification: FoxO-mediated cytokine production regulates diseases

Line 301: …in maintaining…

Line 314: Suggested modification: Collectively, FoxO transcription factors regulate…

Figure 2 legend: …cell types…through FoxO transcription factors.

Line 327: …is involved…

Line 328:  …that the FoxO transcription factors are key regulators…and are implicated…

Figure 2: Is MO indicating macrophages? Usually these are indicated as MΦ. Please include the abbreviation for these cells in the legend.

Author Response

Comments and Suggestions for Authors

This is a Review article on the role of the Forkhead Box O family of transcription factors (FoxOs) mainly focusing on the adaptive and innate immune system. It also aims to summarize the role of FoxOs in oxidative stress-related diseases and potential use as a therapeutic target.  

The authors have gathered a lot of information (past and recent) on the subject, however, reading the manuscript gives the impression that the information is diffuse, not well organized and being repeated. The manuscript needs to be re-organized in clearly defined sections that gather all relevant information, for example everything about DCs and other cell types collected in one section (as in the section 3). In the current version, information mentioned in the introduction about various cell types appear again later in the text.

There are two sections (2 and 3) with the same title “Function of the FoxO family” and so it is not clear how the authors intend to separate this information. Section 3 is the most organized one, and maybe the authors could keep this one as the main part and move all relevant information of the introduction (which can be decreased) and section 2 in that one. The title is about using FoxOs as potential therapeutic targets in oxidative stress-related diseases, however there was no background of which diseases fall into this definition in the introduction.

Response: Thanks for your comments. We added mention of this in Introduction section.

Also, there were no information included related with the therapeutic efforts, and pharmacologic approaches for targeting the FoxO factors.

Response: Thanks for your comments. We added in the Introduction section (line 83).

The role of FoxOs (FoxO1) on Th17 has not been adequately addressed. Here is a paper on the role of FoxO factors in Th17 differentiation that could be helpful: Laine, A. et al. Foxo1 is a T cell-intrinsic inhibitor of the RORgammat-Th17 Program. J. Immunol. 195, 1791-1803 (2015). FoxO factors have also been involved in T cell metabolism, fitness, exhaustion and memory in chronic infections and cancer involving mitochondria and ROS production. (also relevant as part of this IJMS Special Issue ‘Chronic Inflammation and Related Diseases: From Mechanisms to Therapies”). Maybe the authors could add some of this information. Below are some relevant papers that could be discussed.

Staron, M.M. et al. The transcription factor FoxO1 sustains expression of the inhibitory receptor PD-1 and survival of antiviral CD8(+) T cells during chronic infection. Immunity 41, 802-814 (2014).

Scharping, N.E. et al. The tumor microenvironment represses T cell mitochondrial biogenesis to drive intratumoral T cell metabolic insufficiency and dysfunction. Immunity 45, 374-388 (2016).

Zhang, L. et al. Mammalian Target of Rapamycin Complex 2 Controls CD8 T Cell Memory Differentiation in a Foxo1-Dependent Manner. Cell Rep. 14, 1206-1217 (2016).

Odorizzi, P.M., Pauken, K.E., Paley, M.A., Sharpe, A. & Wherry, E.J. Genetic absence of PD-1 promotes accumulation of terminally differentiated exhausted CD8+ T cells. J. Exp. Med. 212, 1125-1137 (2015).

Scheffel, M.J. et al. N-acetyl cysteine protects anti-melanoma cytotoxic T cells from exhaustion induced by rapid expansion via the downmodulation of Foxo1 in an Akt-dependent manner. Cancer Immunol. Immunother. 67, 691-702 (2018).

Response: Thanks for your comments. We added in the 3.6.1. Section (line 331).

Please see below some suggestions for improving the manuscript structure:

  1. Introduction. Should include the general background about FoxO biology, members, structure and general mechanisms of action and activity regulation, not in extensive detail, but briefly by referring to previous reviews, without many details about their cell-specific function and also provide brief information about the oxidative stress-related diseases that will be discussed later in the text. In relation to their structure and activity regulation you can briefly refer to the opportunities opened to target FoxOs therapeutically (e.g. interfering with other proteins that regulate FoxO activity, nuclear transclocation, protein interactions etc…) which will be discussed in more detail later.

Response: Thanks for your comments. We added it.

  1. Function of the FoxO family of transcription factors. Here add the role of FoxOs in the various immune cell types, collecting all the diffuse information from the current version in clearly defined sections.

Response: Thanks for your comments. We added it in the 2 section (line 104).

  1. Current section 3.6 could become section 3 on the role of FoxOs on oxidative stress-related diseases.  Maybe here separate two bigger subsections for autoimmune/inflammatory diseases and cancer/tumor immunotherapy, with subsections on the diseases you want to focus on (e.g. allergic inflammation, SLE, lung inflammation, tumor immunotherapy etc…).

Response: Thanks for your comments. We separate section of 3.6.1. and 3.6.2.

  1. Therapeutic exploitation of FoxO transcription factors. Here you can add current state about pharmacological approaches targeting FoxOs focused on the above discussed diseases. Adding a summarizing table would be very helpful for the readers.

Response: Thanks for your comments. We added the Table 1.

  1. Conclusions. Summary of current state in targeting FoxOs therapeutically (focused on the discussed diseases), future perspectives and challenges.

Response: Thanks for your comments. We added section of conclusion (line 400).

The figures are nice, but I think the second figure can come first with the part mentioning the role of FoxOs in various cell types and then add the figure about FoxOs and ROS in diseases along with that part, including T cell exhaustion and role in tumor immunotherapy/TME if you decide to include it.

Response: Thanks for your comments. We changed Figure 2.

The first paragraph of the conclusions seems a bit odd. These details can also be moved appropriately to relevant section of the main text. Then, the conclusions can start with the second paragraph that you already have.

Response: Thanks for your comments. We moved the first paragraph of the conclusions.

Lines 99-100 and 141: Opposite role of FoxO3 in epithelial versus DCs in IL-6, TNFa production. Maybe discuss that the role of FoxOs may be different or opposite among different cell types (immune vs non-immune cells).

 Response: Thanks for your comments. We added section of 3.2 (line 174).

There is a mistake in Line 64: …CTLA-4-Ig-mediated stimulation… (in ref 26, Dejean et al used CTLA-4-Ig to stimulate B7 receptors on DCs)

Response: Thanks for your comments. We changed (line 79).

There are also some minor corrections for improving the text:

Title: Since FoxO includes a family of many factors (FoxO1, FoxO3…), it would be more appropriate to write as: FoxO transcription factors: Applicability as novel immune cell regulators and therapeutic targets in oxidative stress-related diseases.

Response: Thanks for your comments. We changed title.

Line 12: Forkhead box O transcription factors (FoxOs) play… The same modification is suggested in other places in the text.

Response: Thanks for your comments. We’ve edited our manuscript according to your suggestion and red color the changes (line 13).

Line 15: …by FoxOs…

Response: Thanks for your comments. We changed (line 16).

Lines 16-17: …FoxOs play…responses. They control the…

Response: Thanks for your comments. We changed (line 18).

Line 21: In addition, FoxOs are stimulated…

Response: Thanks for your comments. We changed (line 22).

Line 23: …of FoxOs…

Response: Thanks for your comments. We changed (line 24).

Line 27: …of FoxOs as key molecules…and their role…

Response: Thanks for your comments. We changed (line 28).

Line 29: …FoxOs act as cancer repressors…activity of FoxOs…

Response: Thanks for your comments. We changed (line 30).

Line 30: …FoxOs regulate…

Response: Thanks for your comments. We changed (line 31).

Line 32: …of FoxOs regulation…

Response: Thanks for your comments. We changed (line 33).

Line 37: Forkhead box O transcription factors (FoxOs) are known to play…

Response: Thanks for your comments. We changed.

Line 39: Normally, FoxOs upregulate…

Response: Thanks for your comments. We changed (line 55).

Line 42: …FoxOs are essential…

Response: Thanks for your comments. We changed (line 57).

Line 43: …of FoxOs…

Response: Thanks for your comments. We changed (line 59).

Line 47: …FoxOs are also rendered inactive…

Response: Thanks for your comments. We changed (line 63).

Line 49: …FoxOs play…

Response: Thanks for your comments. We changed (line 65).

Line 53: …stem cells…

Response: Thanks for your comments. We changed (line 69).

Line 55: …homing to lymph nodes…

Response: Thanks for your comments. We changed (line 71).

Line 59: FoxO1 is also… Publication (Dong et al. 2017) needs formatting.

Response: Thanks for your comments. We arrange of ref. 25 (line 75).

Line 61: Remove “function”.

Response: Thanks for your comments. We removed.

Line 72: FoxOs transcriptionally modulate…

Response: Thanks for your comments. We changed (line 95).

Line 75: Sequential phosphorylation of FoxOs by…

Response: Thanks for your comments. We changed (line 97).

Line 79: …of FoxOs, thus leading to their…

Response: Thanks for your comments. We changed (line 101).

Line 82: FoxO proteins are primary regulators of…

Response: Thanks for your comments. We changed (line 109).

Line 83: …by FoxOs…

Response: Thanks for your comments. We changed (line 110).

Line 88: …CTLA-4…

Response: Thanks for your comments. We changed (line 116).

Line 89: …antioxidants.

Response: Thanks for your comments. We changed (line 116).

Section 2 has same title as section 3: “Function of the FoxO family”.

Response: Thanks for your comments. We changed title of section 3.

Line 117: FoxO proteins are…

Response: Thanks for your comments. We changed (line 145).

Line 119: …in the role of FoxOs… FoxO proteins are…

Response: Thanks for your comments. We changed (line 147).

Line 122: Citation reformatting needed to put all refs in one bracket: ...[47, 49] [44, 48, 50].

Response: Thanks for your comments. We arranged (line 150).

Line 126: …main active FoxO protein…

Response: Thanks for your comments. We changed (line 155).

Line 129: Suggested modification: …among the FoxO family members…

Response: Thanks for your comments. We changed (line 157).

Line 147: Macrophages

Response: Thanks for your comments. We changed (line 177).

Line 168: …T cell homeostasis

Response: Thanks for your comments. We changed (line 197).

Line 170: …FoxOs have…

Response: Thanks for your comments. We changed (line 200).

Line 182: I think saying …hampers this balance… would be better.

Response: Thanks for your comments. We changed (line 212).

Line 185: …importance of FoxO proteins in maintaining…

Response: Thanks for your comments. We changed (line 215).

Line 216: …the use of CTLA-4-Ig-mediated stimulation led to…

Response: Thanks for your comments. We changed (line 246).

Line 237: …using an NKp46-Cre mouse model. (remove “that differs from the model”)

Response: Thanks for your comments. We changed (line 267).

Line 239: There is an extra space after [79].

Response: Thanks for your comments. We changed (line 269).

Line 270:…FoxO proteins participate…

Response: Thanks for your comments. We changed (line 302).

Line 284: …TNF-α

Response: Thanks for your comments. We changed (line 316).

Figure 1: Suggested modification: FoxO-mediated cytokine production regulates diseases

Response: Thanks for your comments. We changed (line 322).

Line 301: …in maintaining…

Response: Thanks for your comments. We changed (line 343).

Line 314: Suggested modification: Collectively, FoxO transcription factors regulate…

Response: Thanks for your comments. We modified (line 362).

Figure 2 legend: …cell types…through FoxO transcription factors.

Response: Thanks for your comments. We changed (line 366).

Line 327: …is involved…

Response: Thanks for your comments. We changed (line 361).

Line 328:  …that the FoxO transcription factors are key regulators…and are implicated…

Response: Thanks for your comments. We changed (line 385).

Figure 2: Is MO indicating macrophages? Usually these are indicated as MΦ. Please include the abbreviation for these cells in the legend.

Response: Thanks for your comments. We added in Figure 2 legend.

Round 2

Reviewer 2 Report

The authors revised the manuscript as suggested, but there are new important mistakes that need to be fixed, both major and minor.

Major corrections:

Line 328: Reference 109 is not correct here. The one you have at 114 is the right one. There are more mistakes with references (see further down). If using software for referencing, better convert first to unformatted and then re-format.

Lines 332-333: Please revise the sentences and correct the referencing: Suggested modification: … FoxO1 is considered as an anti-inflammatory control switch directly acting on the Th17 program and deficiency of FoxO1 increases Th17 differentiation while it also results in functionally defective Treg cells [110].

Line 334: Suggested modification, also please correct the reference:…FoxO1 also regulates the expression and function of PD-1, opening new therapeutic options for chronic viral infection or cancer [111].

Line 337: Please correct the reference: [112]

Lines 337-338: Please revise the sentence and correct the reference: Suggested modification: N-Acetyl Cysteine (NAC) antioxidant capacity reduced FoxO1 levels through Akt activation leading to improved tumor control by adoptively transferred antigen-specific TCR-transduced murine T cells [113].

Line 362: Please correct reference 14. (29 and 74 are relevant ones)

Table 1: The references should have numbering as in the main text. b-Catenin is not a pharmacological compound. In reference 123 they tested Wnt-β-catenin signaling inhibitors, PKF and IWR-1.

Minor corrections:

Title: … as novel immune cell regulators...

Line 13: …play an important role… (since FoxOs is a plural form, then the following verb does not take “s”. I noted the same mistake in other places too, but you can also go through to double check.

Line 18: ..play…

Line 19: …control…

Line 28: …as the key molecules…

Line 29: …and their role…

Line 30: …act as cancer repressors…

Line 31: …FoxOs…

Line 45: Suggested modification: ...which regulates…

Line 52: …of FoxOs…

Line 65: …play…

Line 104: …However, disruption…

Line 174: …may have different or opposite functions…

Line 288: …in autoimmunity… or …in autoimmune diseases…

Line 335: …regulates…

Figure 2. Various cell types…

Line 370: …suppresses…

Line 375: …of phosphorylated…

Line 380: …diseases…